microsystems/bioengineering/chemical engineering

*Pseudomonas aeruginosa* PAO1, biofilm growth and eradication, microstructures, caesium chloride, strontium chloride

**Authors for correspondence:**
Jae-Jin Shim
e-mail: jjshim@yu.ac.kr
Chankyu Kang
e-mail: chemnet75@korea.kr

This article has been edited by the Royal Society of Chemistry, including the commissioning, peer review process and editorial aspects up to the point of acceptance.

†These authors contributed equally to this work.

# Physico-chemical characterization of caesium and strontium using fluorescent intensity of bacteria in a microfluidic platform

Changhyun Roh[1,2,†], Thi Toan Nguyen[3,†], Jae-Jin Shim[3] and Chankyu Kang[4]

[1]Decommissioning Technology Research Division, Korea Atomic Energy Research Institute (KAERI), 989-111 Daedukdaero, Yuseong, Daejeon 34057, South Korea
[2]Biotechnology Research Division, Advanced Radiation Technology Institute (ARTI), Korea Atomic Energy Research Institute (KAERI), 29 Geumgu-gil, Jeongeup, Jeonbuk 56212, South Korea
[3]School of Chemical Engineering, Yeungnam University, 280 Daehak-ro, Gyeongsan, Gyeongbuk 38541, South Korea
[4]Office for Government Prime Minister's Secretariat, Service for Promoting Safety of People's Lives, 261 Dasom-ro, Sejong 30107, South Korea

CR, 0000-0002-0542-4828; CK, 0000-0003-0480-3663

Recently, the impact of radioactive caesium (Cs) and strontium (Sr) on human health and the ecosystem has been a major concern due to the use of nuclear energy. However, this study observed changes in green-fluorescent (GFP)-tagged *Pseudomonas aeruginosa* PAO1 biofilms by injecting non-radioactive caesium chloride (CsCl) and strontium chloride ($SrCl_2$) into microstructures embedded in polydimethylsiloxane microfluidic devices, which were used due to their strong toxicity limitations. Four types of microstructures with two different diameters were used in the study. The change of biofilm thickness from fluid velocity and wall shear stress was estimated using computational fluid dynamics and observed throughout the experiment. The effect of pore space became a significant physical factor when the fluid was flowing through the microfluidic devices. As the pore space increased, the biofilm growth increased; therefore, triangular microstructures with the largest pore space showed the best growth of biofilm. Caesium chloride (CsCl) and strontium chloride ($SrCl_2$), less toxic than radioactive caesium (Cs) and

strontium (Sr), completely eradicated the *P. aeruginosa* PAO1 biofilm with low concentrations. The combined effect of toxicity, fluid velocity, wall shear stress and microstructures increased the efficiency of biofilm eradication. These findings on microfluidic chips can help to indirectly predict the impact on human public health and ecosystems without using radioactive chemicals.

## 1. Introduction

The formation of a biofilm on the surface involves very complex and diverse events, among which attachment is the most important aspect of biofilm formation. The complex modification of bacteriophages from planktonic to biofilm means that biofilms can tolerate various external stressors by agglomerating with one another [1]. There are several factors that influence the structure of biofilms, such as substratum characteristics, characteristics of the microorganisms, hydrodynamic conditions and nutrient availability [2–5]. The formation of a biofilm in porous media is important in many environmental and industrial processes, such as bioremediation, oil recovery and wastewater treatment. A typical example of having porous structures is soil, where there are many types of bacteria. Among them, *Pseudomonas aeruginosa* is a common Gram-negative, rod-shaped bacterium that is found in soil, water and skin flora. *Pseudomonas aeruginosa* adapts well to the lifestyle of biofilm, both in the environment and in the course of pathogenesis and results in chronic opportunistic infections due to its strong multidrug resistance [6]. Not only do porous media provide interesting habitats for bacteria, they also have high specific areas and tortuous pore structures that provide extensive grades of several physico-chemical gradients [7]. However, the formation of biofilm structures in complex geometries is still not well understood because of the important roles of various subsurface activities, such as bio-clogging, water transport and various chemical cycles.

Caesium (Cs) is an alkali metal with properties similar to potassium (K), which is an essential element in the growth and development of plants. Strontium (Sr), which reacts vigorously with water, is a soft and alkaline-earth metal with properties similar to calcium (Ca). Both Cs and Sr are taken up by plants and enter the food chain due to the physical and chemical similarities of Cs to K and Sr to Ca [8]. Moreover, the presence of these contaminants adversely affects ecosystem sustainability and contributes to the loss of biodiversity [9]. However, the influence of chemical toxicity on the ecosystem is a relatively undeveloped area compared to bioremediation processes. A few studies reported the toxicity of Cs [10–12] and Sr [13,14] towards plants, mammals and the human body. However, these studies are still lacking in understanding and characterization of the detailed mechanisms of toxicity of Cs and Sr.

Microfluidic platforms for cell cultures and biological applications are currently being carried out with the recent developments in microfluidic devices [15,16]. A precisely controlled environment allows real-time observations of the cell-to-cell and cell-to-extracellular matrix (ECM) interactions, single-cell handling [17], cell culture [18], biofilm formation [19] and chemical toxicity [20]. In addition, microstructures embedded in microfluidic devices enhance mixing efficiency and influence fluid behaviours depending on shape and size [21–23]. These microstructures have also been used as mediators to investigate the effects of fluids on bodily organs or soil [24,25].

The purpose of this study was to investigate the complex interrelations among microorganisms, embedded microstructures and non-radioactive chemicals in a single microfluidic platform. Green-fluorescently (GFP) labelled *Pseudomonas aeruginosa* PAO1 bacteria, mainly found in soil or aquatic habitats, were used as a versatile biological marker to assess the toxicity and microstructural effects of non-radioactive CsCl and $SrCl_2$ by analysing PAO1-GFP biofilm changes. Optically transparent PDMS (polydimethylsiloxane) microfluidic devices with embedded microstructures of various shapes and diameters were used for the real-time monitoring of biofilm growth as well as for the exposure of CsCl and $SrCl_2$ to identify physico-chemical properties of biofilms against non-radioactive chemicals. The toxicity effect of CsCl and $SrCl_2$ was verified by mimicking the various crystal structures of the soil through an *in vitro* system using microfluidic systems. These variously shaped structures may affect flow velocity in nature and may also affect biofilm eradication. Four types of microstructures are embedded in the microfluidic chip: circle (CM), square (SM), hexagon (HM) and triangle (TM). The changes to the PAO1-GFP biofilms by CsCl and $SrCl_2$, which directly allow for the visualization of cell distribution with fluorescent labelling, and the analysis of quantifying biofilm was proportional to fluorescence intensity, were observed by a confocal laser scanning microscopy (CLSM). A microfluidic channel without microstructures was used as a control sample to compare biofilm growth and

eradication with microfluidic channels containing microstructures. In addition to the effects of toxicity of the CsCl and SrCl$_2$ used, the effects of flow velocity and shear stress were also investigated using computational fluid dynamics (CFD). To the best of our knowledge, this study was the first to investigate the effects of non-radioactive Cs and Sr on biofilms using microfluidic devices. Although this study was done only in the laminar flow and confined space as well as a form of non-radioactive salt of Cs and Sr, it could be used as a basic study to indirectly identify the toxicity of radioactive chemicals known to have highly toxic properties [26].

# 2. Material and methods

## 2.1. Materials

CsCl (99.9%) and SrCl$_2$ (greater than 95.0%) were purchased from Sigma Aldrich (St Louis, MO, USA) and Science Company (Lakewood, CO, USA), respectively. Deionized (DI) water was used in the entire experimental procedure to make the appropriate CsCl and SrCl$_2$ concentrations. The bacterial strain used in this study was the *P. aeruginosa* PAO1 wild-type strain obtained from the Korea Atomic Energy Research Institute (Jeongeup, Jeonbuk, South Korea).

The bacterial strain was transformed by electroporation with plasmid pMRP9-1, which stably expresses a green-fluorescent protein (GFP). *Pseudomonas aeruginosa* strain PAO1 with plasmid pMRP9-1 was grown in a Luria–Bertani (LB) medium supplemented with carbenicillin (50 µg ml$^{-1}$) for recombinant selection and retention. *Pseudomonas aeruginosa* containing the GFP plasmid pMRP9-1 was grown on a microfluidic chip and examined by scanning confocal microscopy. The stationary phases used for column chromatography (silica gel 60, 70–230 mesh) and thin layer chromatography (TLC) plates (silica gel 60 F254) were purchased from Merck KGaA (Darmstadt, Germany). The final concentration of bacteria in this study was $1 \times 10^9$ cells ml$^{-1}$.

## 2.2. Microfluidic chip fabrication

Microfluidic chips with various shapes of microstructures were fabricated using standard photoresist-based soft lithography with SU-8 resin [27]. Auto CAD software (AutoDesk, Inc., San Rafel, CA, USA) was used to produce a mask design that was then printed on a transparent film by CAD/Srt Service, Inc. (Brandon, OR, USA). A negative photoresist (SU8 2000, MicroChem Corp, Westborough, MA, USA) was spin-coated onto a 4 inch silicon wafer (Silicon Quest International Inc., Santa Clara, CA, USA). Appropriate UV exposure through the transparent mask design, followed by photoresist development, enabled the formation of a microfluidic device mould with a height of $50 \pm 3$ µm. PDMS (GE RTV 615; elastomer : cross-linker = 10 : 1) was poured onto the wafer and cured at 80°C for 1 h to increase PDMS cross-linking. The microfluidic channel was then peeled off the wafer/photoresist and holes for inlet and outlet ports were made with a 19 gauge punch (Technical Innovation Inc., Brazoria, TX, USA). The PDMS microfluidic channels were combined with a glass slide ($75 \times 50 \times 1$ mm, Ted Pella Inc., Redding, CA, USA) with a hydrophilic surface formed by oxygen plasma (plasma cleaner PDC-326, Harrick Plasma Inc., Ithaca, NY, USA). The bonded PDMS microfluidic channel with the glass slide was placed in an 80°C oven for 45 min. Figure 1*a* shows the actual microfluidic device image used in the experiment. The microfluidic chips contained fabricated channels with a width of $243 \pm 1$ µm and a height of $100 \pm 5$ µm. An aligned row of periodic microstructures was arranged along the centreline of each channel. The micro-fabricated microstructures in this study had diameters of $172 \pm 8.0$ µm (i.e. CM2, SM2, HM2 and TM2) and $132 \pm 6.5$ µm (i.e. CM4, SM4, HM4 and TM4), which were used to elucidate the effects of complex microstructures. Through these fabrication processes, microstructures were formed in the microfluidic devices and the pore space was created. The geometry information of fabricated microfluidic devices is shown in table 1.

## 2.3. Eradication of biofilm using CsCl and SrCl$_2$

CsCl and SrCl$_2$ samples were collected from microfluidic devices prior to injection and after passage through the channels and analysed using inductively coupled plasma mass spectrometry (ICP/MS, analytical instrument 820-MS, Jena, Germany) for sub-ppb to 1 ppm level and inductively coupled plasma optical emission spectrometry (ICP-OES) for 100 ppb to 100 ppm (ICP-OES 8300, Perkin

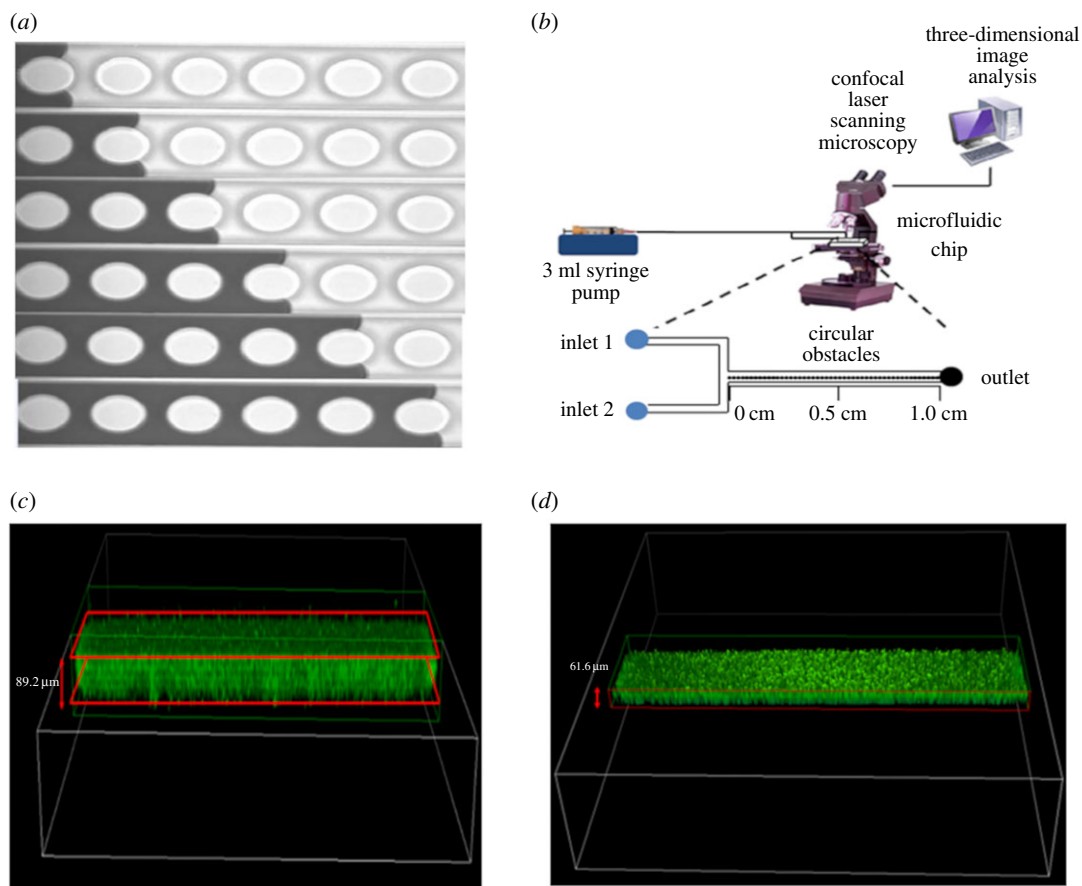

**Figure 1.** (*a*) Flow images recorded for the circular microstructures in microfluidic channels, (*b*) experimental set-up for PAO1 biofilm growth and eradication, (*c*) determination of biofilm thickness by selecting top and bottom, and (*d*) the effect of $SrCl_2$ (0.1 $\mu$M) without microstructures.

Elmer, Waltham, MA, USA), as shown in electronic supplementary material, figure S1(a). Through this process, the concentration used in this experiment and the concentration to be made were verified to provide the reliability of the concentration effect. Detailed apparatus operation and sample preparation methods are well documented in the literature [28].

Figure 1*a* presents the fluid flowing through the microfluidic chip and figure 1*b* shows the experimental apparatus and procedures. No bubbles in the microfluidic channels were detected by a liquid sensor (MLS, Elveflow Plug & Play Microfluidics, Paris, France). A dual syringe pump (Pump 11 Elite Syringe Pump, Harvard Inc., Holliston, MA, USA) was injected into the microfluidic chip using two tygon tubings (0.06″ OD × 0.0200″ ID, Saint-Gobain Corp., Akron, OH, USA). The accuracy of the equipment was 0.5% and the repeatability was 0.5%. LB media and PAO1-GFP were injected into the two inlets at 0.05 $\mu$l min$^{-1}$ for biofilm growth. The formation of the PAO1-GFP biofilm was observed for 5 days. During this period, the microfluidic device was maintained at 37°C in a water bath (VWR International, PA, Radnor, USA) for the cell cultures without additional LB media. After biofilm was maximally increased, the same flow rate of CsCl and $SrCl_2$ was injected into the biofilm-grown microfluidic chip to determine the toxicity of the non-radioactive chemicals. The effects of shear stress, flow rate, local velocity, microchannel geometry and toxic chemicals on biofilm formation and eradication were all considered in this study [29,30].

To analyse the direct measurements of the biofilm formed, three-dimensional volume analysis of the CLSM (Nikon, Tokyo, Japan) was used to select the top and bottom of the biofilms, as shown in figure 1*c*. The thickness of the biofilm was measured at the inlet, middle and outlet portions of the microfluidic channel. In each position, the height and the standard deviation were obtained by measuring at both wall portions and the middle portion. The effect of CsCl and $SrCl_2$ was also measured by the same method. The eradication of PAO1-GFP biofilms by CsCl and $SrCl_2$ in the microfluidic devices monitored by CLSM is shown in figure 1*d*. To analyse the direct measurements of the biofilm formed, three-dimensional volume analysis of the CLSM was used to select the top and bottom of the

**Table 1.** Analysis of the caesium and strontium toxicity against *P. aeruginosa* PA01 biofilm thickness.

| shape of microstructures | pore space (%) | concentration of caesium | | | | concentration of strontium | | | |
| | | 0.1 μM | | 1.0 μM | | 0.01 μM | | 0.1 μM | |
| | | before (μm) | after (μm) | before (μm) | after (μm) | before (μm) | after (μm) | before (μm) | after (μm) |
|---|---|---|---|---|---|---|---|---|---|
| CM2 | 54 | 76.60 | 59.07 | 78.22 | 49.80 | 81.11 | 61.71 | 77.65 | 45.04 |
| CM4 | 70 | 84.48 | 67.12 | 86.46 | 61.24 | 89.35 | 71.77 | 85.93 | 56.14 |
| HM2 | 56 | 78.95 | 59.92 | 82.89 | 51.38 | 85.78 | 63.57 | 82.36 | 45.65 |
| HM4 | 72 | 87.51 | 69.84 | 89.31 | 61.73 | 92.20 | 73.35 | 88.70 | 53.41 |
| SM2 | 41 | 73.68 | 55.13 | 75.56 | 46.37 | 78.45 | 55.43 | 75.06 | 37.17 |
| SM4 | 62 | 80.57 | 61.48 | 84.12 | 58.07 | 87.01 | 64.00 | 83.67 | 45.47 |
| TM2 | 71 | 85.12 | 65.92 | 87.99 | 57.37 | 90.88 | 72.10 | 87.99 | 51.38 |
| TM4 | 81 | 87.52 | 70.88 | 88.74 | 65.73 | 91.63 | 77.48 | 88.21 | 59.36 |
| without microstructures | 100 | 88.03 | 72.15 | 90.16 | 68.64 | 92.17 | 78.28 | 89.20 | 61.57 |

biofilms. Since flow velocity and wall shear stress are known to directly or indirectly affect the thickness of the biofilm, CFD (ANSYS Fluent 14.0, Lebanon, NH, USA) was used to identify the effect [31,32]. For laminar flow, the average wall shear stress ($\tau_{WD}$) is proportional to the dynamic viscosity ($\mu$) and the mean velocity of the fluid ($U_m$), while inversely proportional to the hydraulic diameter ($D_h$) [29], as shown in the following equation:

$$\tau_{WD} = \frac{8\mu U_m}{D_h}.$$

(2.1)

The optimal CsCl and SrCl$_2$ concentrations for the microfluidic devices were predetermined using a Petri dish-based system. After that, various concentrations were injected directly into the PAO1-GFP biofilm, and the range of CsCl and SrCl$_2$ concentrations was finally determined during this process [33]. The difference between the selected highest and lowest concentration was within the range of 100 times. CsCl and SrCl$_2$ solutions were injected for several minutes through the microfluidic device and repeated three times in each case. The fluorescence was stimulated at a 488 nm wavelength, and a filter isolated the wavelength emitted between 505 and 539 nm, which is the green portion of the visible spectrum [34]. The optimal power varies with the objective lens and the type of fluorophore. Images were collected with 400 ms exposure times. Image stacks analysed for each strain at each time point were compiled using ×10 and ×40 objective lenses with a Z-scan to determine the thickness of the PAO1-GFP biofilm by volume measurements. The three-dimensional image files generated by the Z-scan were used to determine the top and bottom of the biofilm and provide detailed information on the actual width, length and depth without complicated procedures. In addition, the actual fluorescence intensity, which indicates the bacterial distributions, was monitored using an intensity surface plot by CLSM (X-axis: position, Y-axis: pixel). During the image processing, the focal-plane depth, laser power and signal gain were maintained at constant values according to the magnification of the lens. In CLSM, the working distance generally varies with the optical design of the system, but typically ranges from hundreds of micrometres to several millimetres [35]. In this study, a working distance of 160–250 µm was applied according to lens magnification. Each experiment was repeated at least three times in order to increase the reliability of the experiment. A new microfluidic channel was used each time to prevent error due to surface quality and contamination of microfluidic channels.

# 3. Results and discussion

The standard curve using ICP/MS equipment (see electronic supplementary material, figure S1(a)) is shown in electronic supplementary material, figure S1(b), where 0.5, 1, 5 and 10 ppb were used. The linear regression correlation was more than 0.99, highlighting accuracy for sample analysis. The concentration of the prepared SrCl$_2$ sample was 5 ppb, and electronic supplementary material, figure S1(c) shows the ICP/MS results. Two samples (inlet: left and outlet: right) showed a similar peak intensity. The analysis was 5.38 and 5.57 ppb, respectively. CsCl also had a similar tendency. All three concentrations of CsCl and SrCl$_2$ used in the experiment were subjected to this process and the results are shown in electronic supplementary material, figure S1(d). This means that the concentration of the manufactured chemical solutions was accurate without a change in concentration as they passed through the microfluidic devices.

Numerous studies have already been conducted on the formation of seeding bacteria into channels for a fixed duration and then supplying nutrients through a known flow rate to form a PAO1 biofilm [36–38]. However, this study shows the effects of microstructures embedded in microchannels in the absence of nutrients. The effects of the complex geometries on biofilm formation are shown in figure 2. In the case of the triangular microstructure, the optimal PAO1-GFP biofilm formed at 72 h and decreased thereafter (figure 2a). This tendency was also observed with the other shapes of microstructures. The increase in biofilms over time was evident and the largest biofilm thickness was found at 72 h. However, the thickness of these biofilms began to decrease after 96 h. Due to the effect of nutrient restriction, the biofilm thickness decreased significantly after 120 h, as compared to 96 h. These results could be applied regardless of the shape of the microstructures. Meanwhile, the four differently shaped embedded microstructures showed different levels of biofilm growth depending on their diameter, as shown in figure 2b. When the size of the microstructure was large, the growth of the biofilm decreased compared to biofilm growth with small microstructures. In this image, square microstructures with a diameter of 132 µm (i.e. SM4) showed not only better biofilm growth, but also spatial uniformity compared to microstructures with a diameter of 172 µm (i.e. SM2). The mean

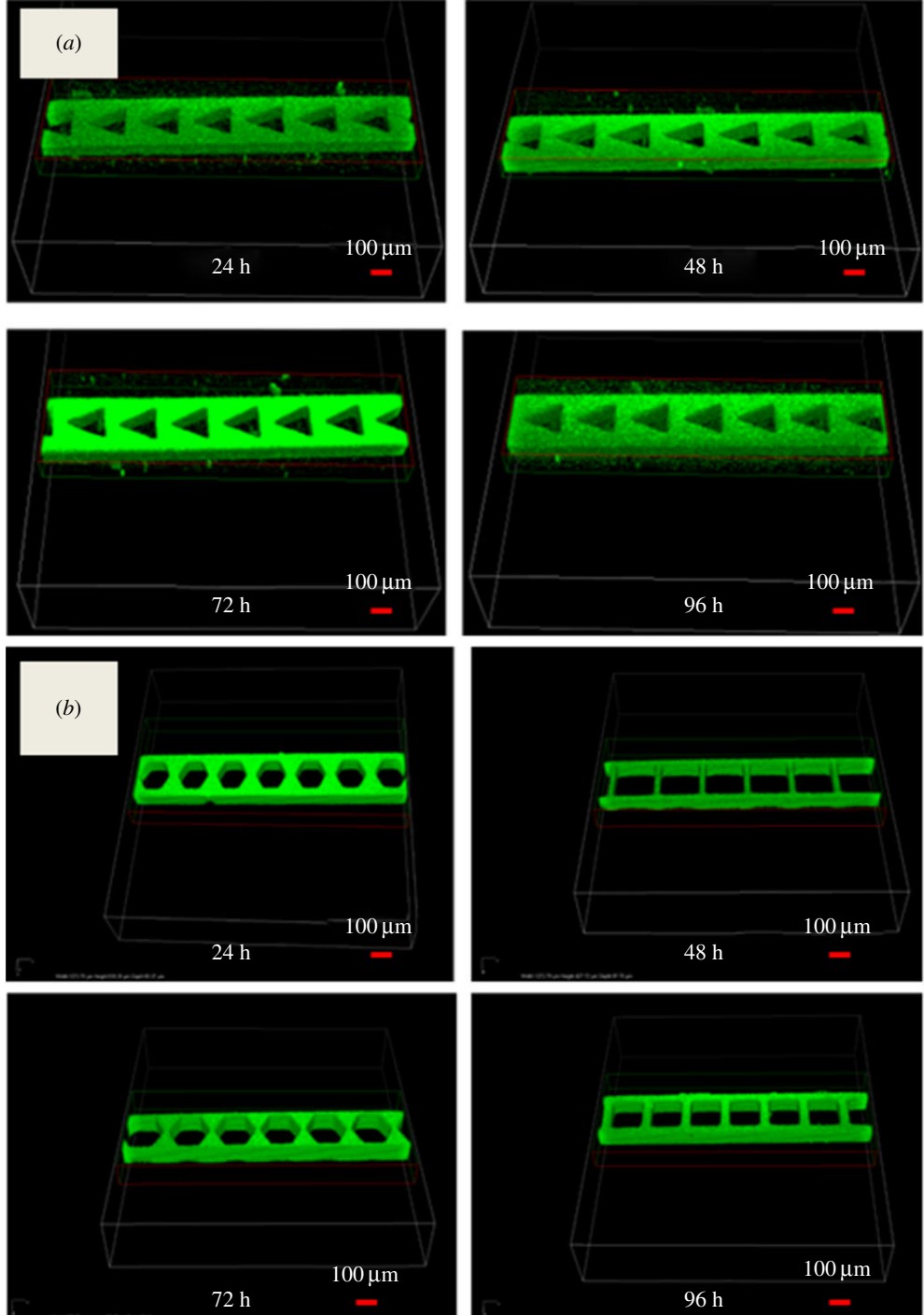

**Figure 2.** (*a*) Growth of biofilm in a microfluidic device with triangular obstacles and (*b*) in microfluidic devices with hexagonal and square obstacles of two different diameters.

difference in biofilm growth between two microstructures after 72 h was 8.56 μm. The mean spatial variations measured at three locations (i.e. channel inlet, middle and outlet) were 0.34 and 0.42 μm in HM2 and HM4, respectively. The uniformity of these biofilms has led to homogeneous bacterial density, possibly caused by microstructures embedded in microchannels, resulting in uniform mixing and diffusion [39].

Figure 3 summarizes the results of images presented in figure 2 for the growth of biofilm due to physical factors, such as the shape and diameter of the microstructures. The absence of microstructures was used as a control sample and compared with the microchannels containing microstructures. The optimal microstructure for the growth of biofilms was the triangular shape,

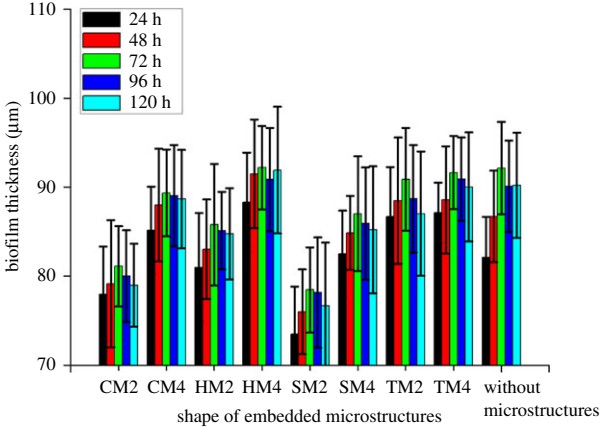

**Figure 3.** Growth of biofilm as functions of time and size of the microstructures.

whereas the worst biofilm formation was observed in the square microstructure. In the absence of microstructures, the spatial variation of biofilm was small and well developed. This means that the microfluidic channel diffusion with high pore space facilitates the diffusion of injected bacteria. Therefore, the biofilm growth was found to be larger as the pore space increased. The effect of pore space through microscale modelling was confirmed to have a similar effect on the growth of uniformly formed biofilm in this experiment [40]. Therefore, biofilm growth in the microfluidic system with a relatively high pore space is easy to achieve and should be considered carefully in microfluidic chip design. This principle also applied equally to the results showing the effects of the size of the microstructures. It has been found that the larger the diameter of the embedded microstructures, the smaller the biofilm thickness, while the smaller the diameter, the greater the biofilm thickness. This increase and decrease in biofilm seems to be related to flow velocity [41]. The distribution of flow velocities on the surface of circular microstructures using CFD simulation is shown in electronic supplementary material, figure S2. The microstructure with a large diameter (i.e. CM2) had a higher flow velocity than the CM4 with a smaller diameter, and the flow velocity increased when the fluid passed through the microstructures. Similar results with soil column are known to have limited cell retention effects at low flow velocities, although the scale is different [31]; therefore, the effect of pore space was relatively important in this experiment.

The eradication of PAO1 biofilm by CsCl and $SrCl_2$ shown in figure 4 was observed at three different concentrations. Toxicity of each chemical was compared by injecting CsCl and $SrCl_2$ into a microfluidic device with a fully grown biofilm. The lowest concentration of CsCl used in this study, 0.1 μM, had the lowest influence on PAO1-GFP and left many residuals on the surface. The bacterial population shown by the fluorescence intensity exhibited strong intensity without CsCl and $SrCl_2$, and decreased with increasing CsCl and $SrCl_2$ concentrations. There were slight residuals of PAO1-GFP biofilms when the concentration of CsCl increased from 0.1 to 1.0 μM, but no residuals were observed when the concentration reached 10.0 μM, due to the strong toxicity of CsCl. As a result, complete eradication was observed when the concentration of CsCl increased 100-fold compared to the initial concentration. However, the toxicity of $SrCl_2$ was much stronger than that of CsCl. The lowest concentration used in this study was 0.01 μM, suggesting that this concentration is still 10 times lower than that of CsCl, but fewer residuals remained due to its toxicity. This means that a small amount of $SrCl_2$ can cause considerable problems when it is exposed to ecosystems. When the concentration of $SrCl_2$ was increased to 0.1 μM, the intensity of the biofilm was weakened, and the residuals finally disappeared at 1.0 μM. It is known that the presence of excess Cs (greater than 200 μM) prevents the absorption of Ca in *Arabidopsis* [29]. It was observed in this study that the concentration of CsCl had a significant effect on the survival of PAO1 bacteria even though the concentration was as low as 1/200. The detailed results of the reduction in PAO1 biofilm by CsCl and $SrCl_2$ are given in table 1. This study showed unexpectedly high toxicity at low concentrations, which would adversely affect human health and the ecosystem.

Both biofilm eradication and biofilm growth were affected by the shape of the embedded microstructures. The chemical properties of CsCl and $SrCl_2$ combined with the physical properties of the microstructures increased the effect of the biofilm. Figure 5 presents the best and

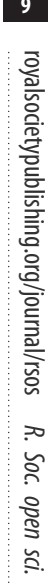

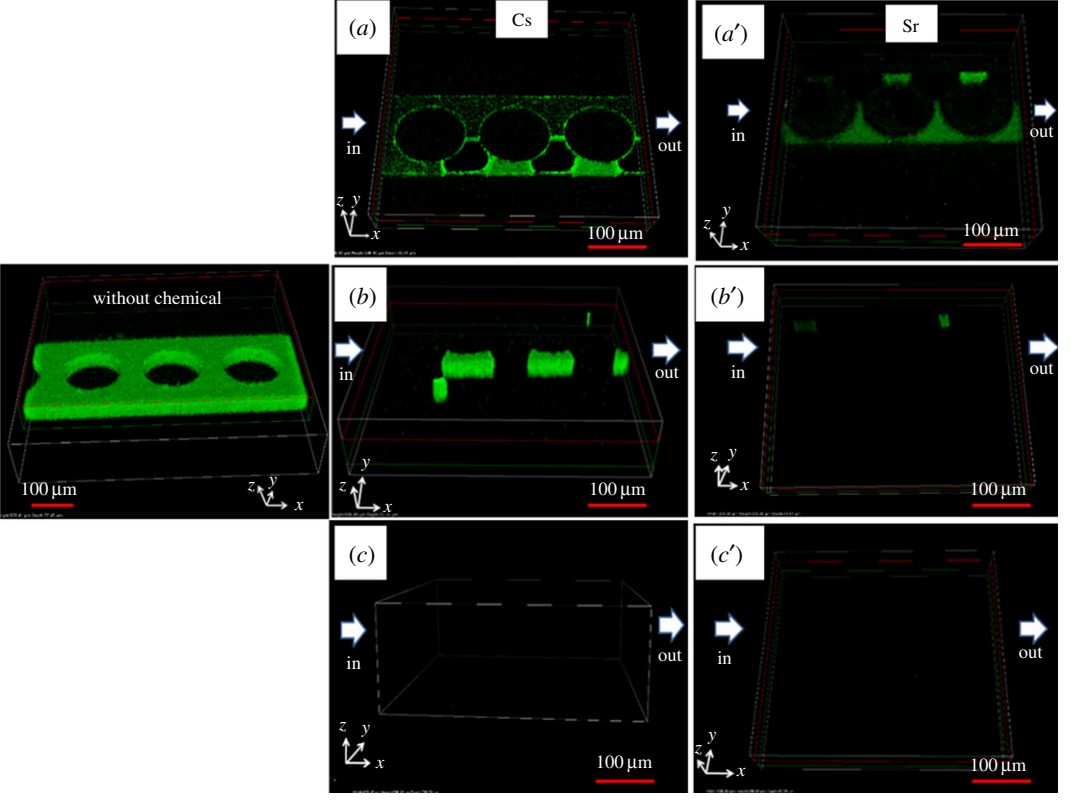

**Figure 4.** Eradication of PAO1-GFP biofilm in microfluidic devices containing circular microstructures using CsCl and SrCl$_2$ at different concentrations: (*a*) 0.1 μM of CsCl and (*a'*) 0.01 μM of SrCl$_2$, (*b*) 1.0 μM of CsCl and (*b'*) 0.1 μM of SrCl$_2$, and (*c*) 10 μM of CsCl and (*c'*) 1.0 μM of SrCl$_2$.

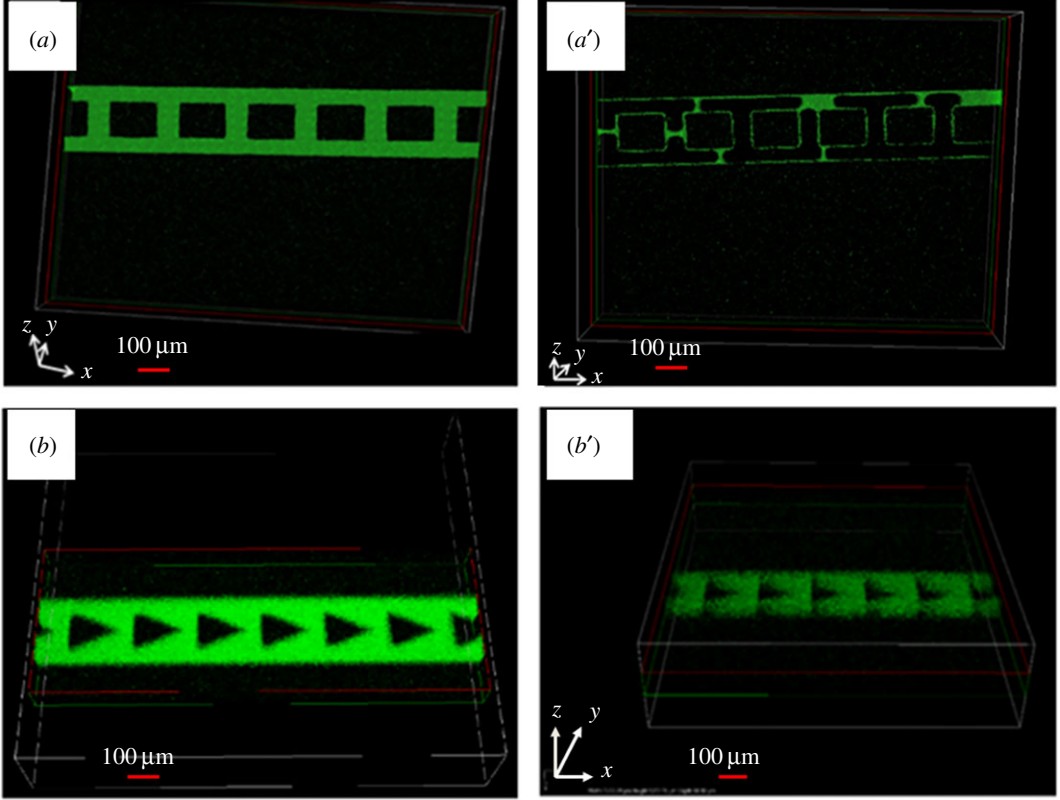

**Figure 5.** Eradication of biofilm in two different microstructures with square and triangular shapes: without SrCl$_2$ (*a,b*) and with 0.01 μM SrCl$_2$ (*a',b'*).

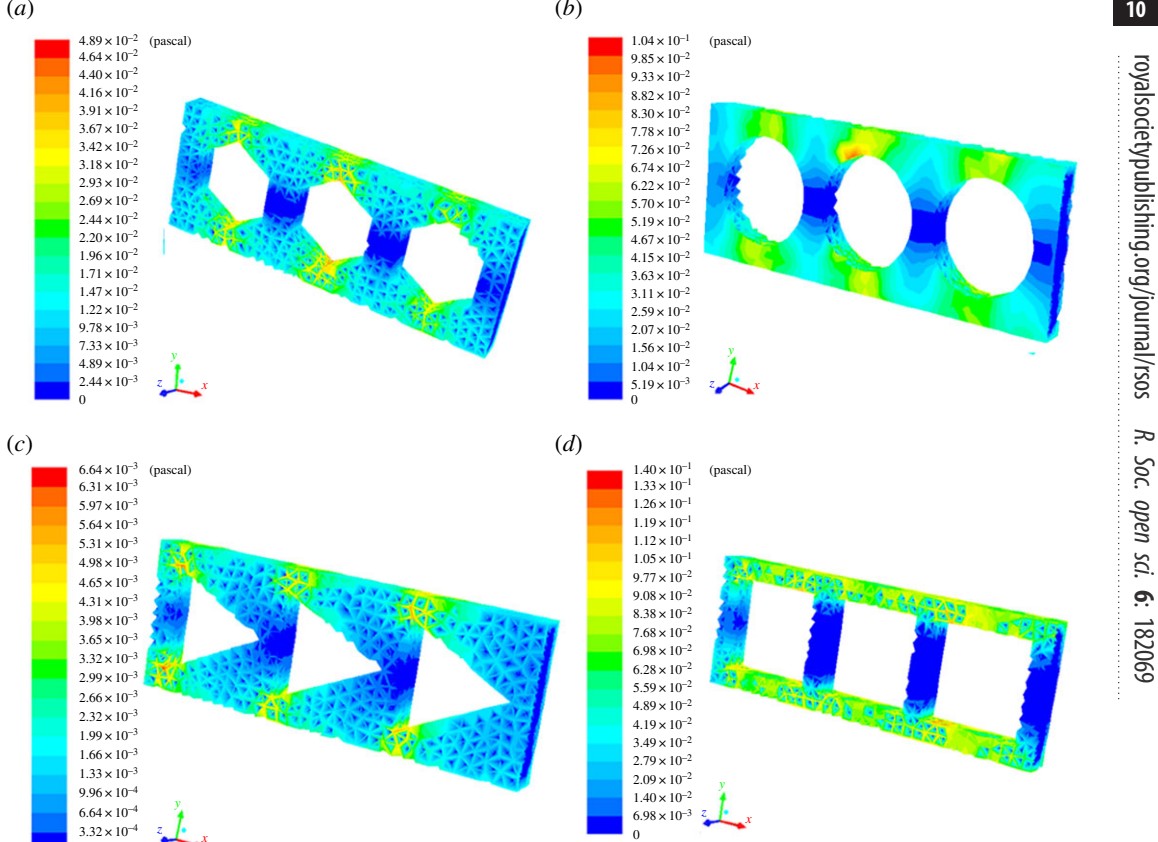

**Figure 6.** Wall shear stress distribution obtained from CFD simulation for microstructures with (*a*) hexagonal, (*b*) circular, (*c*) triangular and (*d*) square microstructures.

worst cases in biofilm eradication using $0.01 \, \mu M$ $SrCl_2$. Among the four types of embedded microstructures, the best eradication was observed in square microstructures, while the triangular microstructures showed the lowest eradication. However, the decrease in biofilm without microstructures was weaker than that with the presence of microstructures. When the same concentration of $SrCl_2$ was injected, the effect on the biofilm was greatly affected by the shape of the microstructures.

In order to investigate the effect of these various microstructures, the effect of wall shear stress, as shown in figure 6, was simulated using CFD analysis. It was known that wall shear stress occurred as the fluid passed through the microstructures, and the biofilm thickness decreased as the strength increased [29]. Among the four types of microstructures, both the greatest wall shear stress and the largest reduction in biofilm thickness, as observed through CLSM, were found in square microstructures. Observations obtained from CFD and these experiments demonstrated the importance of microstructures in removing biofilm.

Figure 7 shows the effects of the size of embedded microstructures on biofilm eradication. The $0.1 \, \mu M$ of CsCl solution was injected into circular microstructures with different diameters and the effects of physical factors were observed. This demonstrated that the effects of the PAO1-GFP biofilm varied according to the diameter of the embedded microstructures. Similar results were also observed with $SrCl_2$. When the diameter of the microstructures embedded in the microfluidic devices was large, it was quite effective in reducing the PAO1-GFP biofilms in combination with the toxicity of CsCl and $SrCl_2$. In the previous studies, it was proved experimentally that the flow velocity was the fastest at $172 \, \mu m$ and that the flow rate was slow at $132 \, \mu m$ [23]. Increased diffusive mass transfer by microstructures and increased wall shear stress due to increased flow velocity resulted in a decrease in biofilm thickness [42,43]. In conclusion, eradication of the PAO1-GFP biofilm was facilitated in microfluidic devices with embedded microstructures with large diameters and high pore space. This result reflects the distinctive features of complex geometries in eradication and biofilm growth.

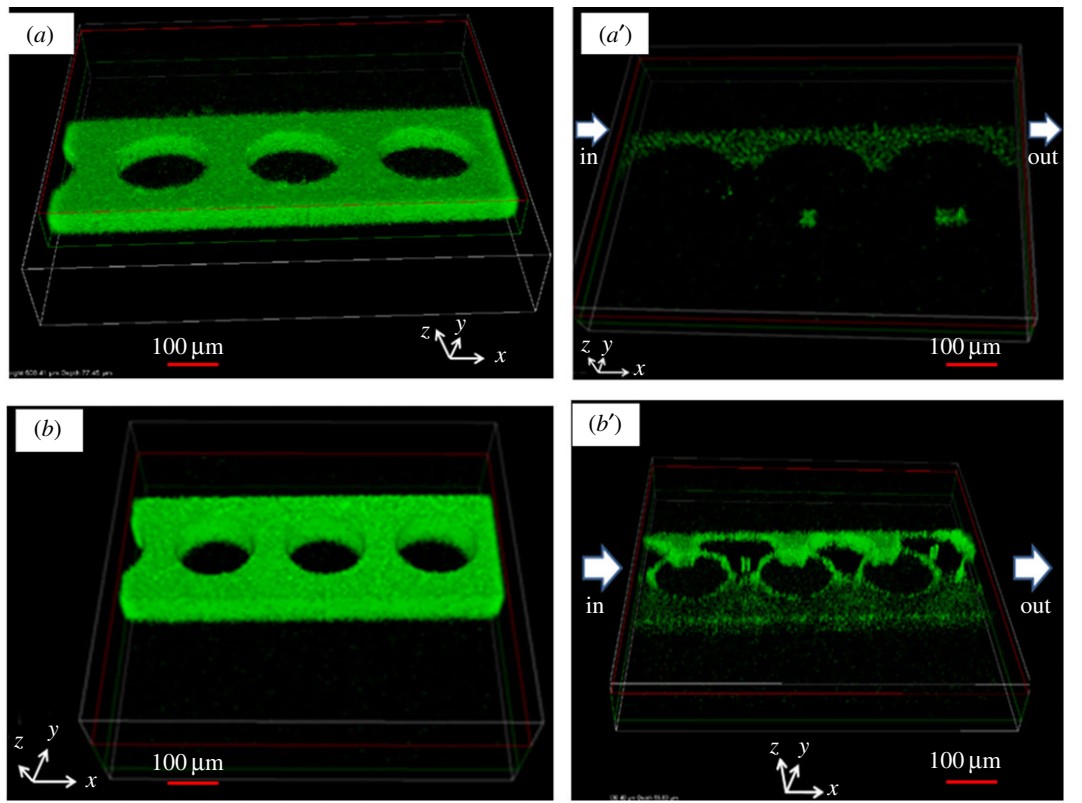

**Figure 7.** Eradication of PAO1 biofilm formed on a microfluidic device with microstructures of different diameters without CsCl (*a,b*) and with 0.1 μM CsCl (*a′,b′*).

## 4. Conclusion

This paper describes the characterization of physical and chemical effects by analysing PAO1-GFP biofilm growth and eradication by low concentrations of CsCl and SrCl$_2$. The microstructures embedded in the microfluidic device were investigated not only for the physical effects of their designs but also for the effects of the two chemicals, which are chemical factors. The changes of biofilm were observed through CLSM using optically transparent microfluidic devices, with embedded microstructures mimicking the structures of soil to perform *in vitro* experiments. The optimal PAO1-GFP biofilm was formed in 72 h regardless of the shape and size of the microstructures, and the biofilm thickness decreased afterward due to nutrient constraints. Among the four differently shaped microstructures, the triangular microstructure showed the best biofilm growth. However, the growth of biofilm without microstructures was larger than that of with microstructures. In addition, as the diameter of the microstructures decreased, the thickness of the biofilm increased. These experimental results were similar to modelling of microscale with porous media. Simulation using CFD showed that the influence of the flow velocity of the embedded microstructures varied with the diameter. However, when the flow velocity was negligibly low, the effect of pore space on the retention of PAO1 bacteria was limited, so pore space was an important factor in the increase in biofilm. The microstructures embedded in the microfluidic devices themselves contributed to the increase in diffusive mass transfer, which is considered to have created a favourable environment for biofilm growth. An analysis of the chemical toxicity showed that SrCl$_2$ at low concentrations exhibited very strong toxicity and had a significant effect on the eradication of PAO1-GFP biofilms. CsCl was less toxic than SrCl$_2$, but surprisingly, PAO1-GFP biofilms were eradicated by low concentration chemicals. The shape of the microstructures embedded in the microfluidic devices, together with the two chemicals used, influenced the PAO1-GFP biofilms. The best biofilm eradication was found in square microstructures, while triangular microstructures had the least effect. The effect of biofilm eradication without microstructures was less than that with microstructures. Analysis by CFD simulations of wall shear stress, which is known to adversely affect biofilm thickness, indicated that the largest wall shear stress was found in square microstructures. Observations of the biofilm thickness using CLSM showed the greatest decrease in square microstructures, probably due to the

greatest effect on wall shear stress. PAO1-GFP biofilm eradication was effective in large diameter embedded microstructures. This indicates that the larger the diameter of the microstructures, the greater the flow velocity and hence the greater the wall shear stress. These results suggest that ecosystems with complex geometries amplify the ripple effect in addition to amplifying the toxicity of CsCl and SrCl$_2$.

Ethics. This research was approved by the Yeungnam University under laboratory operation and safety management regulations. We received written consent from all participants.

Data accessibility. The datasets supporting this article have been uploaded as part of the electronic supplementary material.

Authors' contributions. C.K. and J.-J.S. conceived and designed the experiments; C.R. and T.T.N. performed the experiments; C.R. and T.T.N. analysed the data; all authors wrote the paper; C.K. and J.-J.S. reviewed and edited the manuscript; T.T.N. and C.R. contributed equally to this work.

Competing interests. The authors declare no competing interests.

Funding. This study was supported by the National Research Foundation of Korea (NRF) under the framework of Priority Research Centers Program (NRF-2014R1A6A1031189). This work was also supported by the National Research Foundation of Korea (NRF) funded by the Korean government through the Basic Science Research Program (NRF-2017R1D1A1B06033589) and Nuclear R&D Project (NRF-2018M2A7A1074802).

Acknowledgements. We thank Dr Sangchul Kim of Korea Center for Disease Control for helping us with statistical analysis.

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
