## [Reviewer comments · Royal Society Open Science]

Review History

RSOS-182069.R0 (Original submission)

Review form: Reviewer 1

Is the manuscript scientifically sound in its present form?

Yes

Are the interpretations and conclusions justified by the results?

Yes

Is the language acceptable?

No

Is it clear how to access all supporting data?

Yes

Do you have any ethical concerns with this paper?

No

Have you any concerns about statistical analyses in this paper?

No

Recommendation?

Accept with minor revision (please list in comments)

Comments to the Author(s)

This MS has an interesting story, but the authors should improve English significantly throughout the text. In the abstract, do not use abbreviations without full names.

Review form: Reviewer 2

Is the manuscript scientifically sound in its present form?

Yes

Are the interpretations and conclusions justified by the results?

Yes

Is the language acceptable?

Yes

Is it clear how to access all supporting data?

Yes

Do you have any ethical concerns with this paper?

No

Have you any concerns about statistical analyses in this paper?

No

Recommendation?

Accept with minor revision (please list in comments)

Comments to the Author(s)

The manuscript presents the investigation of the complex interrelations among microorganisms, embedded microstructures and non-radioactive chemicals in a single microfluidic platform. The authors have carried out characterization of physical and chemical effects by analysis of *Pseudomonas aeruginosa* PAO1 PAO1-GFP biofilm growth and eradication by low concentrations of CsCl and SrCl₂. The manuscript is well written, quite interesting and informative. Experimental results support well with the authors conclusion. The manuscript can be accepted after addressing the following minor comments.

1. How did you determine the optimal concentration of CsCl and SrCl₂?
2. While representing the types of embedded microstructures as CM₂, SM₂, HM₂ and TM₂; and CM₄, SM₄, HM₄ and TM₄, what does the labels 2 and 4 represent?
3. Check that the following statement "The thickness of the biofilm was measured at the inlet, middle, and outlet portions of the microfluidic channel. In each position, the height and the standard deviation were obtained by measuring at both wall portions and the middle portion. The effect of CsCl and SrCl₂ was also measured by the same method." is repeated twice in the

same paragraph of the Results and Discussion Section.

4. Why does the triangular microstructure exhibit least eradication?

5. Check the inconsistent usage of SrCl₂ and SrCl₂ and typographical errors throughout the article.

Decision letter (RSOS-182069.R0)

12-Mar-2019

Dear Dr Kang:

Title: Physico-chemical characterization of Cesium and Strontium using fluorescent intensity of bacteria in a microfluidic platform
Manuscript ID: RSOS-182069

Thank you for submitting the above manuscript to Royal Society Open Science. On behalf of the Editors and the Royal Society of Chemistry, I am pleased to inform you that your manuscript will be accepted for publication in Royal Society Open Science subject to minor revision in accordance with the referee suggestions. Please find the reviewers' comments at the end of this email.

The reviewers and handling editors have recommended publication, but also suggest some minor revisions to your manuscript. Therefore, I invite you to respond to the comments and revise your manuscript.

Please also include the following statements alongside the other end statements. As we cannot publish your manuscript without these end statements included, if you feel that a given heading is not relevant to your paper, please nevertheless include the heading and explicitly state that it is not relevant to your work. We have included a screenshot example of the end statements for reference.

- Ethics statement

Please clarify whether you received ethical approval from a local ethics committee to carry out your study. If so please include details of this, including the name of the committee that gave consent in a Research Ethics section after your main text. Please also clarify whether you received informed consent for the participants to participate in the study and state this in your Research Ethics section.

OR

Please clarify whether you obtained the necessary licences and approvals from your institutional animal ethics committee before conducting your research. Please provide details of these licences and approvals in an Animal Ethics section after your main text.

OR

Please clarify whether you obtained the appropriate permissions and licences to conduct the fieldwork detailed in your study. Please provide details of these in your methods section.

- Acknowledgements

Because the schedule for publication is very tight, it is a condition of publication that you submit the revised version of your manuscript before 21-Mar-2019. Please note that the revision deadline

will expire at 00.00am on this date. If you do not think you will be able to meet this date please let me know immediately.

Best wishes,
Dr Laura Smith
Publishing Editor, Journals

RSC Associate Editor:
Comments to the Author:
(There are no comments.)

RSC Subject Editor:
Comments to the Author:
(There are no comments.)

Reviewer comments to Author:
Reviewer: 1

Comments to the Author(s)
This MS has an interesting story, but the authors should improve English significantly throughout the text. In the abstract, do not use abbreviations without full names.

Reviewer: 2

Comments to the Author(s)
The manuscript presents the investigation of the complex interrelations among microorganisms, embedded microstructures and non-radioactive chemicals in a single microfluidic platform. The authors have carried out characterization of physical and chemical effects by analysis of *Pseudomonas aeruginosa* PAO1 PAO1-GFP biofilm growth and eradication by low concentrations of CsCl and SrCl₂. The manuscript is well written, quite interesting and informative. Experimental results support well with the authors conclusion. The manuscript can be accepted after addressing the following minor comments.

1. How did you determine the optimal concentration of CsCl and SrCl₂?
2. While representing the types of embedded microstructures as CM2, SM2, HM2 and TM2; and CM4, SM4, HM4 and TM4, what does the labels 2 and 4 represent?
3. Check that the following statement "The thickness of the biofilm was measured at the inlet, middle, and outlet portions of the microfluidic channel. In each position, the height and the standard deviation were obtained by measuring at both wall portions and the middle portion. The effect of CsCl and SrCl₂ was also measured by the same method." is repeated twice in the same paragraph of the Results and Discussion Section.
4. Why does the triangular microstructure exhibit least eradication?
5. Check the inconsistent usage of SrCl₂ and SrCl₂ and typographical errors throughout the article.

Author's Response to Decision Letter for (RSOS-182069.R0)

See Appendix A.

Decision letter (RSOS-182069.R1)

05-Apr-2019

Dear Dr Kang:

Title: Physico-chemical characterization of Cesium and Strontium using fluorescent intensity of bacteria in a microfluidic platform

Manuscript ID: RSOS-182069.R1

It is a pleasure to accept your manuscript in its current form for publication in Royal Society Open Science. The chemistry content of Royal Society Open Science is published in collaboration with the Royal Society of Chemistry.

RSC Associate Editor
Comments to the Author:
(There are no comments.)

Reviewer(s)' Comments to Author:

Appendix A

Responding Sheet

Reviewer #1

Q1) This MS has an interesting story, but the authors should improve English significantly throughout the text. In the abstract, do not use abbreviations without full names.

Response) According to your comment, we modified our mistake to do not use abbreviations in the abstract section, and it was also corrected by a company specialized in English proofing ([Editage.com](https://www.editage.com), Job code: YNURS_119, Green color).

Reviewer #2

The manuscript presents the investigation of the complex interrelations among microorganisms, embedded microstructures and non-radioactive chemicals in a single microfluidic platform. The authors have carried out characterization of physical and chemical effects by analysis of *Pseudomonas aeruginosa* PAO1 PAO1-GFP biofilm growth and eradication by low concentrations of CsCl and SrCl₂. The manuscript is well written, quite interesting and informative. Experimental results support well with the authors conclusion. The manuscript can be accepted after addressing the following minor comments.

Q1) How did you determine the optimal concentration of CsCl and SrCl₂?

Response) Methods for determining the optimal concentrations of CsCl and SrCl₂ are given in page 6 (lines 201-204). However, References were added for introducing detailed processes (i.e. Petri dish method) how to determine the optimal concentrations of of CsCl and SrCl₂.

Page 6 (lines 201-204): “The optimal CsCl and SrCl₂ concentrations for the microfluidic devices were pre-determined using a Petri dish-based system. After that, various concentrations were injected directly into the PAO1-GFP biofilm, and the range of CsCl and SrCl₂ concentrations was finally determined during this process [33]”

Q2) While representing the types of embedded microstructures as CM2, SM2, HM2 and TM2; and CM4, SM4, HM4 and TM4, what does the labels 2 and 4 represent?

Response) Labels 2 and 4 are arbitrary numbers and there is a difference in diameter. Detailed information is provided on page 4 (Line 144-146).

Page 4 (Line 144-146): “The micro-fabricated microstructures in this study had diameters of $172\pm 8.0\mu\text{m}$ (i.e. CM2, SM2, HM2, and TM2) and $132\pm 6.5\mu\text{m}$ (i.e. CM4, SM4, HM4, and TM4)”

Q3) Check that the following statement “The thickness of the biofilm was measured at the inlet, middle, and outlet portions of the microfluidic channel. In each position, the height and the standard deviation were obtained by measuring at both wall portions and the middle portion. The effect of CsCl and SrCl₂ was also measured by the same method.” is repeated twice in the same paragraph of the Results and Discussion Section.

Response) We thank you for pointing out our mistakes and corrected the problems (Page 5, Line 178-185).

Q4) Why does the triangular microstructure exhibit least eradication?

Response) It is an important part of our manuscript. We confirmed that the triangular microstructure exhibited the worst effect among the four different shapes of microstructures in biofilm eradication using experimental and numerical methods (i.e. CFD). The average wall shear stress (τ_{WD}) shown in Eq. (1) is affected by the flow velocity. It is known that the increase in shear stress will promote biofilm detachment because high flow rates are preferred at all times to reduce the buildup of bacterial biofilms [1]. In the case of the triangular microstructure, the flow velocity was observed to be the slowest among the four microstructures [2] and the average wall shear stress (τ_{WD}) was proved to be the smallest through CFD analysis.

Reference)

1. Moreira J, Simões, Melo, LF, Mergulhão. 2014 The combined effects of shear stress and mass transfer on the balance between biofilm and suspended cell dynamics. *Desalination and water treatment* **53**, 3348-3354.
2. Kang C, Roh C, Overfelt RA. 2014 Pressure-driven deformation with soft polydimethylsiloxane (PDMS) by a regular syringe pump: challenge to the classical fluid dynamics by comparison of experimental and theoretical results. *RSC Adv.* **4**, 3102.

Q5) Check the inconsistent usage of SrCl₂ and SrCl₂ and typographical errors throughout the article.

Response) We modified our mistake (Page 5, Line 155, 174).